# Cascaded Aggregation Convolution Network for Salient Grain Pests Detection

**DOI:** 10.3390/insects15070557

**Published:** 2024-07-22

**Authors:** Junwei Yu, Shihao Chen, Nan Liu, Fupin Zhai, Quan Pan

**Affiliations:** 1Key Laboratory of Grain Information Processing and Control (Henan University of Technology), Ministry of Education, Zhengzhou 450001, China; chenshihao13@163.com (S.C.); zhaifupin618@126.com (F.Z.); 2Henan Key Laboratory of Grain Photoelectric Detection and Control, Henan University of Technology, Zhengzhou 450001, China; quanpan@nwpu.edu.cn; 3Basis Department, PLA Information Engineering University, Zhengzhou 450001, China; liunan526@126.com; 4School of Automation, Northwestern Polytechnical University, Xi’an 710129, China

**Keywords:** visual attention mechanism, salient object detection, feature aggregation, feature enhancement, cascaded atrous convolution

## Abstract

**Simple Summary:**

Infestations of pests in grain storage can have a significant impact on both the quantity and quality of stored grains. Drawing inspiration from the detection abilities of humans and birds in identifying pests, we present an innovative deep learning solution designed for the detection and management of pests in stored grains. Specifically focusing on the detection of small grain pests within cluttered backgrounds, we propose a cascaded feature aggregation convolution network. Our approach outperforms existing models in terms of both trainable parameters and detection accuracy, as evidenced by experiments conducted on our newly introduced GrainPest dataset as well as publicly available datasets. By sharing our dataset and refining our model’s architecture, we aim to advance the field of research in grain pest detection and the classification of stored grains based on pest density. This study is expected to contribute to the reduction of economic losses caused by storage pests and to enhance food security measures.

**Abstract:**

Pest infestation poses significant threats to grain storage due to pests’ behaviors of feeding, respiration, excretion, and reproduction. Efficient pest detection and control are essential to mitigate these risks. However, accurate detection of small grain pests remains challenging due to their small size, high variability, low contrast, and cluttered background. Salient pest detection focuses on the visual features that stand out, improving the accuracy of pest identification in complex environments. Drawing inspiration from the rapid pest recognition abilities of humans and birds, we propose a novel Cascaded Aggregation Convolution Network (CACNet) for pest detection and control in stored grain. Our approach aims to improve detection accuracy by employing a reverse cascade feature aggregation network that imitates the visual attention mechanism in humans when observing and focusing on objects of interest. The CACNet uses VGG16 as the backbone network and incorporates two key operations, namely feature enhancement and feature aggregation. These operations merge the high-level semantic information and low-level positional information of salient objects, enabling accurate segmentation of small-scale grain pests. We have curated the GrainPest dataset, comprising 500 images showcasing zero to five or more pests in grains. Leveraging this dataset and the MSRA-B dataset, we validated our method’s efficacy, achieving a structure S-measure of 91.9%, and 90.9%, and a weighted F-measure of 76.4%, and 91.0%, respectively. Our approach significantly surpasses the traditional saliency detection methods and other state-of-the-art salient object detection models based on deep learning. This technology shows great potential for pest detection and assessing the severity of pest infestation based on pest density in grain storage facilities. It also holds promise for the prevention and control of pests in agriculture and forestry.

## 1. Introduction

Grains (cereals, oilseeds, and legumes) are the major source of food for both humans and domestic animals. Grain pests can inflict significant damage on stored grains through activities such as feeding, breathing, excretion, reproduction, webbing, and other behaviors. Insect infestation is the leading major factor for postharvest losses of grains during storage [1]. In accordance with the Chinese standard GB/T 29890-2013: Technical Criterion for Grain and Oil-seeds Storage, the presence and severity of grain pests dictate the requirement for implementing control measures such as low-temperature storage, chemical fumigation, and other applicable methods. Stored grain pests can cause substantial post-harvest losses, ranging from approximately 9% in developed countries to potentially exceeding 20% in developing countries [2]. Insect infestation will result in the loss of stored grains both quantitatively and qualitatively, and then affect the nutritional values and marketability of the subsequent foodstuff [3]. Considering the growing global population and the slow growth of food production, the Food and Agriculture Organization (FAO) predicts that the world may face food scarcity in the coming decades [4]. To reduce the major grains postharvest loss in the storage stage, we can take some proper storage methods and intelligent approaches for pest infestation identification and control. With the development of computer vision [5] and emerging deep learning techniques [6], modern approaches based on image processing and recognition can provide rapid, economic, and precise solutions for grain pest identification and detection. 

The popular detection methods of grain pests include manual sampling [7], acoustic detection [8], and computer vision [9]. Manual sampling involves labor-intensive efforts to probe specified locations in the granary, drawing 1 kg of grains for screening pests through sieves. Inspectors visually assess the presence and quantity of pests using their naked eyes. Despite the fact that inspectors have good eyesight and good sense of subtilizing, manual inspection methods are labor-intensive, time-consuming, and subjective [10]. Acoustic detection methods monitor the activities of grain pests with the moving and feeding sound. The effectiveness of acoustic methods depends on understanding the relationship between acoustic waves and pest distribution. The acoustic receiver is also expensive and sensitive to background noise [11]. Advancements in computer vision have led to the integration of various technologies, such as optical instrumentation, near-infrared spectroscopy, X-ray imaging, electromagnetic sensing, image processing, and machine learning, for internal and external grain pest detection and recognition [3]. The USDA (United States Department of Agriculture) and FGIS (Federal Grain Inspection Service) utilize visual reference images for insect infestation and grain grading [12]. Li et al. designed a multi-scale pyramid network with both classification and box regression subnets to detect the common stored-grain insects [13]. Shi et al. proposed an RPN-based convolutional neural network to predict the classification of eight common stored grain insects [14]. Chen et al. [15] introduced an automated pest detection and counting system to address the limitations of dataset context and pest trap-based methods in grain pest detection. The proposed automatic system and YOLOv4 model achieved a mean average precision (mAP) of 97.55%, meeting the practical accuracy requirements for detecting and counting granary pests.

Visual saliency detection [16] refers to the computational process of identifying the most visually prominent regions in an image or video, which are likely to capture human attention. The goal of visual saliency detection is to highlight areas or objects that are perceptually distinct from their surroundings, based on features such as color, texture, shape, or contrast. Coincidentally, humans and birds can identify pests in grains at a glance with the mechanism of visual saliency detection. In the context of our research, salient object detection (SOD) plays a crucial role in identifying and highlighting regions in images that potentially contain grain pests, making grain pest detection more efficient and effective in complex environments. While many solutions based on both traditional image processing algorithms [17] and deep learning models [18] have been proposed in recent years for SOD, some issues remain open for real-world applications such as grain pest detection [19]. Motivated by the visual systems of humans and birds, we aim to address the more complex real-world scenario of grain pest detection. 

To address the above issues, the main contributions and novelties of this paper can be summarized as follows:

(1) GrainPest: A challenging and benchmark dataset for salient grain pest detection. This dataset has the following characteristics: a diversity of grain pests, a high proportion of small-sized objects, the presence of non-salient objects, and pixel-level annotation.

(2) CACNet: A novel one-branch model for saliency detection based on a reverse cascaded feature aggregation convolution network is proposed. We have designed a cascaded atrous convolution module to increase the receptive field and enhance multi-scale feature representations for small targets.

(3) Experiments: We conducted comparative experiments to evaluate the performance of the proposed CACNet by comparing it with both traditional visual saliency detection methods and several state-of-the-art deep learning models on GrainPest and MSRA-B datasets. Quantitative and qualitative results demonstrate that CACNet achieves high detection accuracy, especially for small salient objects.

## 2. Materials and Methods

### 2.1. Cascaded Aggregation Convolution Network

Convolutional Neural Networks (CNNs) can accurately segment images into distinct areas of interest by utilizing convolutional layers to extract features and learn spatial patterns. However, due to the presence of pooling and stride operations, the spatial resolution of these feature maps is usually reduced. This reduction poses challenges in generating visual saliency maps at the original image size using conventional network structures, consequently affecting the accuracy of grain pest saliency detection. To address this issue, we propose a cascaded atrous convolution approach that mimics the visual attention mechanism of humans, involving glancing, searching, and focusing. As shown in Figure 1, we introduce a U-like network called the Cascaded Aggregation Convolution Network (CACNet). CACNet employs cascaded atrous convolution to expand the receptive field of convolutional layers, effectively resolving the challenge of limited spatial resolution in feature maps encountered in conventional convolutional networks. This helps in capturing large-scale contextual information and improving the performance of small pest detection and segmentation. Additionally, we apply reverse cascade feature enhancement and feature aggregation techniques to produce a visual saliency map with high precision.

In the coding process, we utilize well-established image-processing backbone networks, specifically the VGG16 and ResNet50. These networks have demonstrated robust capabilities in representing images and come with pre-trained parameters derived from extensive datasets like ImageNet, eliminating the need for training models from scratch. The backbone network is organized into five stages, each consisting of various components, including 2D convolutions, ReLU activation functions, and average pooling layers. In the lower layers, the image resolution remains higher, resulting in convolution outputs with more detailed edge and positional information. Conversely, the higher layers exhibit an increase in the number of image channels, a decrease in resolution caused by pooling operations, and convolution outputs that incorporate more profound semantic information for classification and segmentation.

In the decoding process, we employ a reverse cascade feature aggregation structure, as shown in the lower half of Figure 1. Directly upsampling or deconvolving deep features would result in a segmented image lacking fine details. Hence, it is essential to aggregate the output results from different stages, considering both high-level semantic information (stage 3 to stage 5) and low-level structural information (stage 1 to stage 2). To increase the receptive field, the output features from layer 3 to layer 5 of the backbone undergo Cascaded Atrous Convolution (CAC) modules. In order to obtain a saliency map that has the same size as the original image, a reverse feature fusion process is applied. The feature fusion process encompasses several upsampling, convolution, and concatenation operations. The reverse cascade feature aggregation involves two primary operations: feature enhancement and feature aggregation. Feature enhancement involves mapping the convolution results from higher layers back to the corresponding layers in the lower level, followed by enhancing this feature map through bitwise multiplication. On the other hand, feature aggregation combines the mapped results from higher layers with the enhanced features of the lower-level layer to create a more comprehensive and informative feature combination. Finally, a 1 × 1 convolution is performed on the final aggregated features to generate a visual saliency map.

The results obtained from the reverse cascade aggregation process are compared with the ground truth to evaluate their performance using the cross-entropy loss function. Since determining pixel membership within the salient target is the main objective of saliency detection, the resulting saliency probability maps can be easily converted into binary features. Hence, we employ the binary cross-entropy loss function in this context. In order to simplify the network architecture, we decide to evaluate the loss solely based on the aggregated features of the last layer in the reverse cascade, diverging from methods such as the Cascaded Partial Decoder (CPD) that incorporate both low-level structural features and high-level semantic features in the loss calculation. Additionally, for the specific task of grain pest detection, our primary focus lies in detecting and quantifying salient pest targets, rendering the detailed delineation of target edges unnecessary. As a result, we do not incorporate an edge-preserving loss function like the Pyramid Feature Attention Network (PFA).

### 2.2. Cascaded Atrous Convolution for Receptive Field

Enhancing the detection ability of small objects requires a dual approach, involving the development of multi-scale feature representations for small targets and the incorporation of contextual information fusion techniques. Grain pests, due to their small size and ability to camouflage and mimic the color of grain kernels, pose a significant challenge for grain pest detection. In contrast, human observers excel in this task by quickly assimilating global and local image information through glancing, searching, and focusing actions. As a result, humans consistently achieve high rates of detection and recognition in identifying grain pests. To replicate the visual processes of human observation, specifically glancing, searching, and focusing, and enhance the detection of grain pest targets, we propose a novel solution: the cascaded atrous convolution module. This module is designed to facilitate the multiscale representation of targets and the fusion of contextual information.

In Figure 2, our module begins by performing a non-linear combination on the channel dimension of the features from the backbone network using a 1 × 1 convolution. These features serve as the foundation for subsequent atrous convolutions and feature aggregation. The operation of 1 × 1 convolution can not only reduce model parameters but also increase the expressive power of the network. We incorporate three sets of atrous (dilated) convolution to increase the receptive field of the convolutional layer and enhance the multiscale feature representation, specifically for small targets. To mitigate grid effects, we employ a combination of ternary primes, such as 3, 5, 7, as dilation rates, which have proven to be effective. These three atrous convolution layers are sequentially concatenated and then aggregated with other branches. Finally, the aggregated features are subjected to element-wise addition with the residual results from the first convolutional layer, followed by an activation function to produce the final output. This mechanism emulates the human visual attention process from coarse to fine details, thereby expanding the model’s receptive field while reducing the model parameters.

### 2.3. Evaluation Metrics

We employ the commonly used evaluation metrics, including the Structure-measure (*Sm*), Mean Absolute Error (*MAE*), mean E-measure (*Em*), and weighted F-measure (*Fm*) to assess the performance of our salient object detection models. To compute these metrics, we utilize the MATLAB toolbox provided by Fan [20]. 

The S-measure (structure measure) takes into account both the object-aware ($S_{o}$) similarity and the region-aware ($S_{r}$) contrast between the predicted saliency map and the ground truth map. It provides a single score that reflects the overall quality of the saliency map, considering both local and global consistency.
(1)$$S_{m}=\alpha \times S_{o}+(1-\alpha )\times S_{r}$$

Mean Absolute Error (*MAE*) is used to evaluate the pixel-level error between the generated saliency map *M* and the ground truth *G*. *W* and *H* denote the width and height of the image, with:(2)$$MAE=\frac{1}{W\times H}\sum_{x=1}^{W}\sum_{y=1}^{H}\left\|M(x,y)-G(x,y)\right\|$$

E-measure (Enhanced-alignment measure) uses the alignment matrix ϕs to capture the pixel-level matching and image-level statistics of the predicted saliency map.
(3)$$E_{m}=\frac{1}{W\times Q}\sum_{i=1}^{W}\sum_{j=1}^{H}\phi s(i,j)$$

Precision (*P*) refers to the accuracy of the algorithm which is the percentage of salient pixels correctly assigned. While recall (*R*) refers to the relationship of detected salient pixels to the ground truth salient pixels. For a given saliency map, we convert it to a binary mask *M* using a varying threshold from 0 to 255. The precision and recall can be computed by comparing the binary mask (*M*) of the saliency map with its ground truth (*G*) as described in:(4)$$P=\frac{\left|M\cap G\right|}{M}$$
(5)$$R=\frac{\left|M\cap G\right|}{G}$$

F-measure, as a weighted summed average of precision and recall, has non-negative weights and its evaluation results are more reliable, and it is calculated as follows:(6)$$F_{\beta }=\frac{(1+\beta^{2})P\times R}{\beta^{2}P+R}$$
where, *β*_2_ = 0.3 is a suggested threshold by previous work. *β*_2_ is computed across the thresholds, and *F*_*β*_max_ represents the maximum overlap between precision and recall. If *F_β_* score is closer to 1, the overlap between the saliency map and the ground-truth is larger.

## 3. Experiments and Results

### 3.1. Dataset

Deep neural networks play an important role in acquiring knowledge from extensive images in many computer vision tasks. Consequently, the quality of the dataset becomes a determining factor in the level of knowledge that can be acquired by the model. Previous studies have revealed that deep learning-based salient object detection models experience significant performance degradation when detecting small targets. In addition to the model’s structure, the choice of the datasets also serves as a critical factor influencing model performance. Many saliency detection datasets, including DUTS-TE [21], DUTS-TR [21], ECSSD [22], HKU-IS [23], and MSRA-B [24] presume the presence of a single salient object in an image. Moreover, several methods rely on center bias to detect the salient object at the image center. However, these biases in data selection can reduce the model’s generalization and performance when applied in complex real-world scenarios.

The SOC (Salient objects in clutter) [20] dataset acknowledges that images composed of cluttered scenes, such as landscapes and textures, typically lack salient objects. To address this, the dataset includes approximately 50% more images with non-salient objects. On the other hand, other datasets often overlook the non-salient objects and primarily focus on medium or large objects. Experimental results indicate that existing saliency detection algorithms, such as PoolNet [25], experience a significant decline in performance when images with non-salient objects are introduced. Furthermore, inconsistencies in the distribution of object scales between training and test sets, as observed in DTUS-TR and DTUS-TE, also impact the performance of trained models. Therefore, it is necessary to construct a dataset composed of small grain pests or pure grain backgrounds without grain pests.

Overall, our dataset comprises 500 images, specifically curated for evaluation purposes, with a focus on small salient objects. Figure 3 illustrates exemplars from the GrainPest dataset, along with their corresponding ground-truth binary masks, which we have labeled to facilitate the evaluation of saliency detection methods.

The sample images and corresponding ground truth saliency maps are illustrated in Figure 3. Existing salient object detection (SOD) models and datasets usually assume that there are one or two big salient objects in the image. The performance of these models will decrease significantly when the image contains many small objects or just non-salient objects. We have to face more challenges as described in Figure 3 from column (a) to (e). Firstly, the image background consists of various grains, including wheat, rice, corn, and unshelled paddy. Secondly, a variety of pests including red flour beetle (*Tribolium castaneum Herbst*), rice weevil (*Sitophilus zeamais Motschulsky*), corn borer larva, wheat moth, and sawtoothed grain beetle (*Oryzaephilus surnamensis*) are parasitized in the grains. Thirdly, the image background is highly cluttered due to the activities of pests. Fourthly, there are always more than two salient objects in each image. Finally, in a lot of cases, the grains are not infected with pests and the image is pure background with non-salient objects. Moreover, most of the pests are as small as grains and they are more easily overlooked than the big and salient objects. Consequently, grain pest detection is a challenging task that requires a novel dataset and network architecture based deep learning. 

We have collected a novel dataset, GrainPest, consisting of 500 images. Most of the images contain one or more small grain pests such as weevils, saw-toothed beetles, and moths in stored grain. For the construction of this dataset, we captured some images in the controlled laboratory environment and searched for additional images from Insect Images https://www.insectimages.org/ (accessed on 10 December 2022), which is a project led by the University of Georgia. To take the non-salient objects into account, we included 50 pure background images with rice, wheat, and corn. Additionally, we provided ground truth annotations in the form of binary masks for the salient grain pests.

Since small objects are often more challenging to detect due to their low resolution and limited visual features, accurately detecting them is an important research topic in computer vision. The COCO (Common Objects in Context) dataset defines small objects as those with a bounding box area of less than 32 × 32 pixels. We evaluate the size of salient objects with the ratio of their pixels, and the ratio can be calculated using Formula (7). In (1), *W* and *H* are the width and height of the image, and *pix*(*x*,*y*) = 1 represents the pixel of the ground truth.
(7)$$R=\frac{\sum_{x=1}^{W}\sum_{y=1}^{H}\left(pix(x,y)=1\right)}{W\times H}$$

Salient object size can be categorized into four levels based on the percentage (*R*) of the object’s pixels in the image: H1 (*R* ≤ 10%), H2 (10% < *R* ≤ 20%), H3 (20% < *R* ≤ 30%), and H4 (*R* > 30%). These levels correspond to different object sizes, with H1 denoting the absence of salient objects or small objects, H2 representing medium objects, H3 denoting medium-large objects, and H4 representing large objects. The scale information for commonly used saliency datasets is presented in Table 1 and depicted in Figure 4. Notably, the GrainPest dataset primarily consists of small target images, accounting for 57.4% of the dataset. Additionally, 16% of the images in the GrainPest do not contain salient objects, while only a mere 0.8% of the images contain large targets. As such, the GrainPest dataset represents a typical dataset primarily focused on small objects.

### 3.2. Experimental Setup

We assessed the performance of our proposed model on two datasets: GrainPest and MSRA-B. The GrainPest dataset, tailor-made for saliency detection related to grain pests, comprises 500 images meticulously annotated by hand. In contrast, the MSRA-B dataset consists of 5000 images encompassing diverse content categories, including natural scenes, animals, indoor settings, outdoor environments, and more. The datasets are split into training and testing sets in a ratio of 8:2.

Experiments are conducted on Pytorch with Python 3.7 as the programming language. VGG16 [26] is used as the backbone. The learning rate starts at 1 × 10^−4^ and the batch size is set to 1 during the training process. The training time is about 70 min for 30 epochs (early-stop strategy) on dataset GrainPest. In other words, it takes about 78.6 s each epoch for every 400 images. The running time is measured on the platform of Intel Core i7-8700 CPU @3.20GHz × 6 (Intel, Santa Clara, CA, USA) and GeForce GTX TITAN X (NVIDIA, Santa Clara, CA, USA) equipped with Ubuntu 18.04.5LTS.

### 3.3. Compared Results

In order to assess the effectiveness of the CACNet model in saliency detection, a comprehensive evaluation was conducted. The performance of CACNet was compared against traditional saliency detection methods, such as GBVS [27], CA [28] and RARE [29], as well as five state-of-the-art deep learning approaches, namely PFA [30], DHS [31], DSS [32], CPD [33], U2Net [34]. Table 2 presents the quantitative evaluation results, including the trainable parameters, as well as four evaluation metrics on two datasets. The best three results are highlighted in red, blue, and green font, respectively. Notably, on the GrainPest dataset, CACNet outperforms other models in nearly all four evaluation metrics. Similarly, on the MSRA-B dataset, CACNet and CPD demonstrate outstanding performance.

To demonstrate the effectiveness of the CACNet model, we present sample results in Figure 5, comparing its performance with that of state-of-the-art methods. As shown, CACNet exhibits superior capabilities in dealing with grain insect images that are characterized by small sizes and cluttered backgrounds. It excels in accurately segmenting grain pests from the grain background, offering a more effective strategy for the detection and control of pests in stored grain.

### 3.4. Ablation Study

To evaluate the effectiveness of our cascaded atrous convolution module, we conducted an ablation study by constructing two distinct models. The first model adhered to the standard CACNet architecture, while the second model, denoted as CAC-NoRFB, omitted the Receptive Field Block (RFB) of the cascaded atrous convolution module. The results, presented in the bottom two rows of Table 2, demonstrate our evaluation of these models on the GrainPest and MSRA-B datasets using four performance metrics. These evaluations highlight the exceptional performance of our method on both datasets, emphasizing the importance of a cascaded atrous convolution module in saliency detection. 

## 4. Discussion

### 4.1. Traditional Solutions

Since the 1990s, both domestic and international research efforts have been committed to the advancement of image recognition technology for detecting and counting grain insects. Notably, the U.S. Federal Grain Inspection Service pioneered the use of visual reference images in their pest infection and grain grading inspection system since 1997. In prior research, Ridgway et al. [35] employed machine vision techniques to detect wheat pests, such as sawflies. Neethirajan et al. [7] conducted an evaluation of advancements in research encompassing sound detection, image recognition, and infrared sensors for stored-product insect detection. Some conventional approaches to grain insect detection and recognition, as outlined by [36], have centered on fundamental feature extraction, including image color, edges, and textures, followed by grain insect localization and detection. Alternatively, certain methodologies involve grain pests counting through region growing and connected component labelling algorithms, while others employ neural networks and support vector machines, leveraging various image features to achieve class recognition of grain pests. However, traditional techniques face challenges in practical grain storage settings due to the variety of grain pest species, their small size, different shapes, occlusion, and various grain storage conditions [37].

Before the era of deep learning, there were quite a few visual methods can get the saliency map using local or global contrast techniques. The resulting saliency map highlights the regions that stand out from the background. Itti and Koch [38] proposed a method which uses a bottom-up approach to compute the saliency map by combining several feature maps, such as color, intensity, and orientation, using a center-surround operation. Bernhard et al. [27] constructed a graph to model the image, where nodes represent image pixels and edges capture the relationships between pixels. The Graph-based Visual Saliency (GBVS) method combines multiple visual cues to accurately identify salient regions in images. Paper [28] introduced a context-aware saliency model (CA) based on four psychological principles to detect the salient objects that are most likely to attract human attention. Literature [39] computes the saliency maps using the topological structure of Boolean maps. The BMS model is efficient for both eye tracking and salient object detection. In [40], the color saliency map and regional stability are combined to detect the small target. Riche et al. [29] introduced a novel saliency prediction model named RARE2012 for identifying regions of interest within an image based on their spatial rarity at multiple scales. 

In addition to the above-mentioned methods based on the spatial domain, many scholars try to detect salient objects in the frequency domain. Hou et al. [41] point out that humans are more sensitive to the changing part of the image. They use a spectral residual (SR) model to extract the salient features of an image by computing the residual of the image’s Fourier spectrum after removing the low-frequency components. The residual image is then binarized to obtain the saliency map. According to [42], the phase spectrum of an image contains more detailed information than the amplitude spectrum. They propose a method called PFT for image reconstruction using solely the phase information. In a similar vein, [43] introduces a saliency detection model based on the amplitude spectrum of the quaternion Fourier transform, taking into account human visual sensitivity. Another saliency detection model called HFT is presented in [44], utilizing the hypercomplex Fourier transform of the color image. By analyzing the amplitude spectrum at different scales, HFT is capable of detecting salient regions of various sizes. In [45], a frequency-tuned (FT) model is introduced, aiming to generate a saliency map at full resolution while preserving most of the frequency information of the image, consequently achieving better segmentation. Additionally, in [46], a new contrast measure is devised for each block in the spectral domain, allowing for the detection of salient regions based on both local and global contrast.

In our experiments, we compared our designed CACNet with three traditional solutions, as presented in Table 2. Traditional methods rely on manually crafted features, which may result in poor detection performance due to the small size of pest targets and cluttered backgrounds in pest images. When compared to deep learning approaches, traditional methods have inherent limitations.

### 4.2. Solutions in the Deep Learning Era

Significant advancements in object detection have been observed in recent years, owing to the development of deep learning techniques [47] and their potent representation learning abilities. Among these techniques, convolutional neural networks (CNNs) [48] have been widely employed in various computer vision tasks such as image classification, object detection, and image segmentation. CNNs, particularly deeper architectures, possess the ability to extract high-level information, enabling effective detection of salient objects. 

There are several studies that have explored the use of object detection algorithms for detecting stored grain pests. These studies typically utilize deep learning-based object detection models, such as Faster R-CNN and YOLO [15], to identify and locate pests in storage facilities. To effectively train the object detection model, a substantial number of images containing both healthy and infested grains is required. These models are designed to learn and identify visual patterns indicative of pest presence, such as discoloration or the physical appearance of the pests. Once trained, the model can be employed to automatically detect and accurately localize pests within new images or video frames.

While significant progress has been made in object detection based on deep learning, there is relatively limited research on pixel-wise segmentation of pests or the application of visual saliency detection methods for detecting grain pests. However, the need for such research is crucial due to the challenges associated with accurately identifying and representing pests within storage images. The ability to perform precise pixel-level segmentation or leverage visual saliency for segmentation is essential for obtaining detailed insights into the extent of infestation, distinguishing pests from other objects, and facilitating proper control strategies. Overcoming these challenges through dedicated research efforts can significantly enhance the efficiency of pest management and contribute to minimizing grain storage losses.

Salient object detection (SOD) aims to identify the focal objects within an image that attract the most attention and subsequently extract their pixel-level silhouettes [20]. The survey discusses various deep learning-based techniques for salient object detection, including fully convolutional networks (FCNs), recurrent neural networks (RNNs), and generative adversarial networks (GANs). 

Over the past few decades, numerous SOD methods have been proposed, including multi-layer perceptron-based methods, fully convolutional network-based methods and hybrid network-based methods, and generative adversarial network-based methods [20,25]. Recent advancements in deep convolutional neural networks have established new state-of-the-art performance in salient object detection. Hou et al. introduce a DSS [32] network for saliency detection, short connections, which incorporates short connections within the deep neural network architecture to leverage both low-level and high-level information. Liu [31] proposed the DHS network to enhance the accuracy of salient object detection by capturing hierarchical features and contextual information. Another deep learning model, CPD [33], utilizes a cascaded partial decoder network to efficiently and accurately identify salient objects in images. Qin et al. [34] introduce U2Net, a nested U-structure designed for salient object detection. U2Net aims to capture complex details and features by delving deeper into the network architecture, facilitating accurate saliency detection. Furthermore, edge and boundary cues are utilized to further refine the saliency map [49]. Attention mechanisms [50] are employed to enhance features, contributing to their effectiveness in saliency detection.

The comparative experiments involving CACNet illustrate the superior performance of deep learning algorithms over traditional solutions. CACNet, comprising reverse cascade feature aggregation structure, enhances various metrics on both datasets compared to the SOTA deep learning models. CACNet achieves top performance in metrics such as MAE, S-Measure, and F-Measure, while also having the smallest model size and trainable parameters. Furthermore, CACNet exhibits the lowest computational complexity in terms of FLOPs among the tested models.

## 5. Conclusions

In response to the challenges posed by cluttered backgrounds and small pest sizes in grain images, we propose a novel Cascaded Aggregation Convolution Network (CACNet) for the saliency detection of grain pests. Drawing inspiration from attention mechanisms observed in human and avian vision, our approach introduces a cascaded atrous convolution module to increase the receptive field of the model. Experimental results indicate the superior performance of the proposed CACNet compared with the state-of-the-art models. By simplifying the architecture and controlling the number of channels for feature aggregation, our model effectively reduces the number of parameters and FLOPs, making it well-suited for application in the grain storage industry. Furthermore, we demonstrate our commitment to advancing this field by releasing a benchmark dataset specifically collected for the saliency detection of pests in stored grain. This study presents a dataset benchmark and an effective model for grain pest detection and segmentation. These contributions lay the foundation for subsequent tasks such as insect species recognition, grain pest density estimation, pest control decisions, and more. To enhance the system’s future applications, we are contemplating the integration of specialized sensors to ensure consistent performance across various environmental conditions.

## Figures and Tables

**Figure 1 insects-15-00557-f001:**
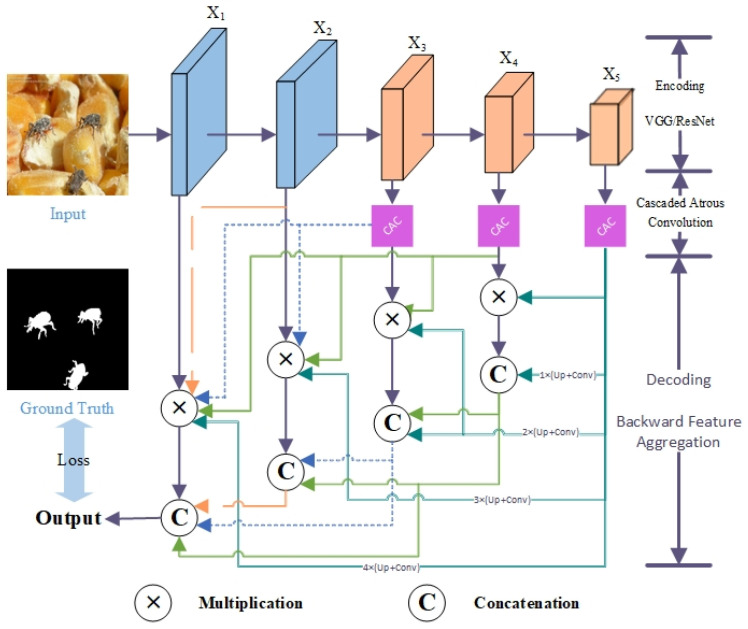
The architecture of the proposed Cascaded Aggregation Convolution Network (CACNet). The encoding process uses a backbone of VGG or ResNet, which is divided into 5 stages. The decoding process involves two operations of feature enhancement (Multiplication) and feature aggregation (Concatenation) to aggregate the output results of different layers.

**Figure 2 insects-15-00557-f002:**
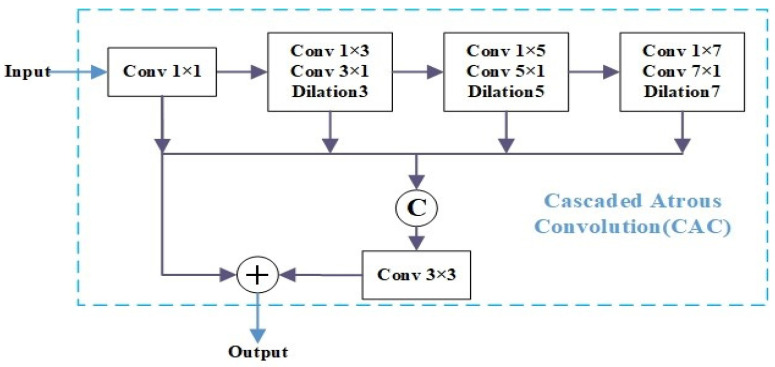
Cascaded atrous convolution module to increase the receptive field. Three atrous convolutions are linked in a cascaded manner to progressively expand the receptive field. Subsequently, the output features from each atrous convolution are concatenated in parallel for extracting enriched representations of the input data.

**Figure 3 insects-15-00557-f003:**
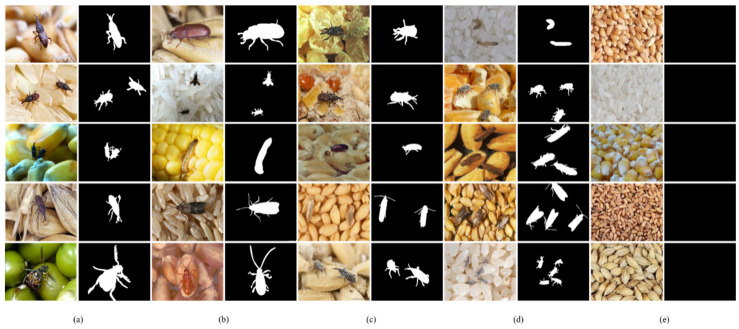
Sample images and challenges in saliency detection of stored-grain pests. The challenges described in each column are as follows: (**a**) variety of grains such as wheat, corn, rice, etc.; (**b**) diverse grain pests; (**c**) cluttered backgrounds; (**d**,**e**) showing pest number variations from zero to more than five.

**Figure 4 insects-15-00557-f004:**
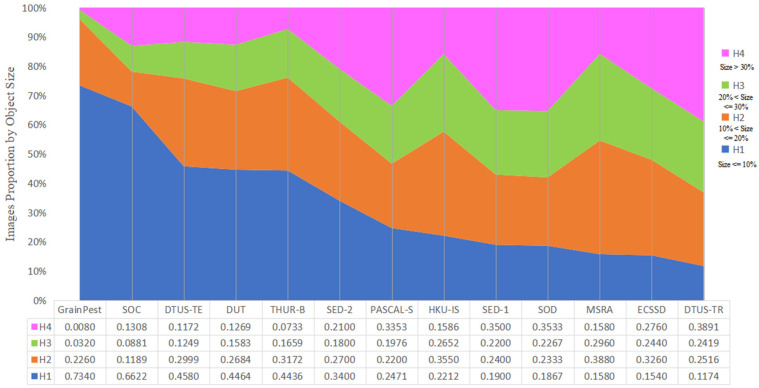
Size distribution of salient objects in common saliency detection datasets. Object size level is characterized by the proportion of salient object pixels to the total image pixels.

**Figure 5 insects-15-00557-f005:**
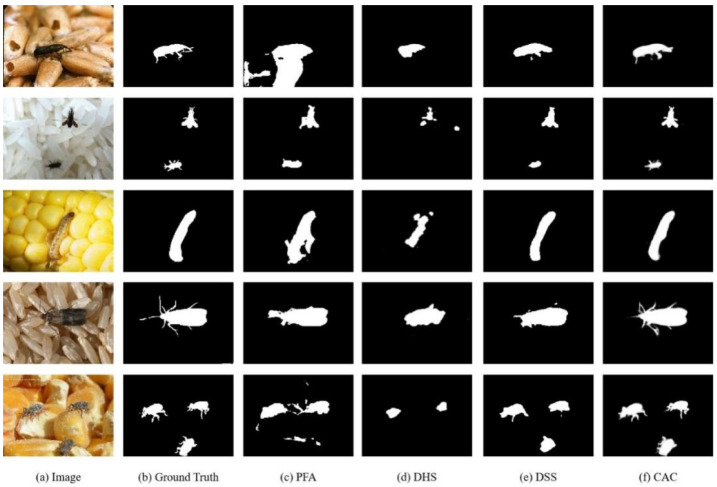
Qualitative comparison results with three SOTA saliency detection models: (**a**) Original Image, (**b**) Ground Truth, (**c**) PFA, (**d**) DHS, (**e**) DSS, (**f**) CACNet.

**Table 1 insects-15-00557-t001:** Dataset summary: image counts and object size distribution in popular saliency detection datasets. H1 to H4 indicate the object size levels of Small, Medium, Medium-Large, and Large Object.

Dataset	Images	H1	H2	H3	H4
DTUS-TE	5019	45.8%	29.99%	12.49%	11.72%
DTUS-TR	10,556	11.74%	25.16%	24.19%	38.91%
ECSSD	1000	15.4%	32.6%	24.4%	27.6%
HKU-IS	4445	22.12%	35.5%	26.52%	15.86%
MSRA-B	5000	14.24%	38.0%	28.52%	19.24%
SOC	3600	66.22%	11.89%	8.81%	13.08%
GrainPest	500	73.4%	22.6%	3.2%	0.8%

**Table 2 insects-15-00557-t002:** Quantitative evaluation results on dataset GrainPest and MSRA-B with traditional saliency detection methods and 5 SOTA deep learning models. ↑ indicates larger is better and ↓ indicates smaller is better. The results in bold indicate the best.

Methods	Params (Mb)↓	GrainPest				MSRA-B			
		S_m_↑	MAE↓	E_m_↑	F_m_↑	S_m_↑	MAE↓	E_m_↑	F_m_↑
GBVS	-	0.625	0.190	0.585	0.508	0.658	0.227	0.536	0.615
CA	-	0.643	0.186	0.603	0.509	0.613	0.250	0.529	0.548
RARE	-	0.702	0.127	0.627	0.533	0.619	0.220	0.501	0.581
PFA	65.6	0.749	0.109	0.822	0.583	0.854	0.058	0.911	0.834
DHS	375.1	0.819	0.031	0.889	0.671	0.872	0.050	0.927	0.881
DSS	249.0	0.875	0.022	0.951	0.720	0.882	0.044	**0.931**	0.885
CPD	183.0	0.912	0.018	**0.958**	0.756	0.905	**0.039**	**0.931**	0.900
U^2^Net	176.3	0.899	0.024	0.912	0.742	0.902	0.048	0.918	0.896
CAC-NoRFB	64.7	0.798	0.064	0.830	0.589	0.890	0.051	0.913	0.887
**CACNet**	**16.0**	**0.919**	**0.017**	0.949	**0.764**	**0.909**	0.044	0.927	**0.910**

## Data Availability

The datasets generated and analyzed during the current study are available from the corresponding author on reasonable request.

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
