# Peer review of "Cascaded Aggregation Convolution Network for Salient Grain Pests Detection"

_insects, 2024, doi:10.3390/insects15070557_

Round 1

Reviewer 1 Report

Comments and Suggestions for Authors

This manuscript is interesting and timely and describes a pest detection system for grain, however I would suggest some improvements:

Simple summary: In my opinion it is very clear, no further suggestions from my side

Abstract: in my opinion abstract should be improved. It should be more detailed. Plese improve and clarify the aim (what is different from the state of the art?), please improve the part of materials and methods because I think should be a little bit more detailed, please add the main results I think few main data are required, please add at the end a very brief opportunity of application of this technology in farms/food industry.

Keywords: please check that keywords do not repeat the words of the title. I think that the keywords should be rewritten.

Introduction: When you talk about the damages to grains I would consider talking a little bit about grains from organic farming systems vs grains from conventional farming systems. Concerning the manual sampling, did you find in the literature data about the precision of this method (maybe expressed as percentage)? The same for acustic. Please add in the paragraph from line 82 to line 96 references or maybe move this part to materials an methods (please see below). The aim is clear

Related works: Personally I would prefer to transform this section entirely to a Discussion section, comparing the literature also to the results obtained in this trial. Maybe more general information should be moved to the introduction instead of the discussion.

Materials and methods: line 235, I would move Figure 1 to Materials and methods, maybe it is a more appropriate location with respect to the introduction. Why 500 image? Please justify this number. Is there a difference between the photos from the lab and thje ones from www.insectimages.org from the point of you of your detection systems? There is typing error LINE 297 "szie". Please clarify the innovation of your casceded aggregaton with respect to an ordinary convolution network. To me the difference from cascade aggregation to cascade atrous is not clear. Which is the main goal of this second step? Please include in Materials and Methods a detailed description of the experiments. The result section should only include results.  Evaluation metrics should also be moved to materials and methods in my opinion.

Results: With my suggested modifications the results section is too small. I think that all the information on your dataset could be moved to results section (maybe including a few more data if you can) in order to make this section more complete.

Discussion: it is completely missing. As I said before I would suggest to transform related works in discussion adding a comparison between the literature and the findings of this trial.

Conclusions: Will your dataset and system be usefull fot farmers and food industry? What practical benefits can your research provide? Would it be more important in organic agriculture where damages can potentially be more consistent? Would you suggest special sensors for the future application of your system? Please clarify.

Thank you in advance for taking my suggestions into accoount and good luck for the publication of your manuscript.

Reviewer 2 Report

Comments and Suggestions for Authors

The authors applied a reverse cascaded aggregation convolution network (CACNet) to generate high-resolution saliency map. They also created a new image dataset GrainPest consisting of grain pests. Their experimentation using CACNet model on this and other datasets outperformed other salient object detection models. This model seems novel and is worth publishing. My comments follow:

The keyword "salient" needs a clear definition. Is it visually prominent?

Since this is an entomology journal, it would be beneficial for readers to know to which species/development stage or groups of species of insects and crops the images used belong.

Round 2

Reviewer 1 Report

Comments and Suggestions for Authors

I sincerely thank the authors for their patience in addressing my comments. I find now the manuscript improved and suitable for publication in this revised version. However please check your reference manager because the references are missing in my pdf ("Error! Reference source not found").

Thank you again and congratulations 

Author Response

Comments: I sincerely thank the authors for their patience in addressing my comments. I find now the manuscript improved and suitable for publication in this revised version. However please check your reference manager because the references are missing in my pdf ("Error! Reference source not found").

Responce: Thank you for your valuable feedback. We have carefully addressed your comments and made the necessary revisions to ensure the accuracy and completeness of the references in the manuscript. We have reorganized and managed the references using cross-referencing to ensure that each reference corresponds correctly to the associated resource. We appreciate your thorough review and are pleased that the manuscript is now improved and suitable for publication in its revised version.